# Macrophage-Immunomodulatory Actions of Bovine Whey Protein Isolate, Glycomacropeptide, and Their In Vitro and In Vivo Digests

**DOI:** 10.3390/nu15234942

**Published:** 2023-11-28

**Authors:** Wyatt Olsen, Ningjian Liang, David C. Dallas

**Affiliations:** 1Department of Food Science & Technology, Oregon State University, Corvallis, OR 97331, USA; wyatt.olsen@oregonstate.edu; 2Nutrition Program, College of Health, Oregon State University, Corvallis, OR 97331, USA; liangn@oregonstate.edu

**Keywords:** caseinmacropeptide, caseinomacropeptide, milk, immunity, Crohn’s disease, ulcerative colitis

## Abstract

Whey protein isolate (WPI) consists of an array of proteins and peptides obtained as a byproduct of the cheesemaking process. Research suggests that WPI, along with its peptides such as glycomacropeptide (GMP), possesses immunomodulatory properties. These properties hold potential for alleviating the adverse effects of inflammatory conditions such as inflammatory bowel disease. Although promising, the immunoregulatory properties of the digested forms of WPI and GMP—those most likely to interact with the gut immune system—remain under-investigated. To address this knowledge gap, the current study examined the effects of in vitro-digested WPI and GMP, in vivo-digested WPI, and undigested WPI and GMP on the secretion of pro-inflammatory cytokines (TNF-α and IL-1β) in lipopolysaccharide-stimulated macrophage-like cells. Our results indicate that digested WPI and GMP reduced the expression of TNF-α and IL-1β, two pro-inflammatory cytokines. Whole WPI had no effect on TNF-α but reduced IL-1β levels. In contrast, in vivo-digested WPI reduced TNF-α but increased IL-1β. Undigested GMP, on the other hand, increased the secretion of both cytokines. These results demonstrate that digestion greatly modifies the effects of WPI and GMP on macrophages and suggest that digested WPI and GMP could help mitigate gastrointestinal inflammation. Further clinical studies are necessary to determine the biological relevance of WPI and GMP digestion products within the gut and their capacity to influence gut inflammation.

## 1. Introduction

Whey protein isolate (WPI) is derived from cheesemaking and consists of an array of proteins and peptides. WPI is a popular nutritional supplement due to its complete amino acid profile, abundance of branched-chain amino acids, and high digestibility [1]. Bovine milk-derived WPI contains hundreds of proteins, including β-lactoglobulin, α-lactalbumin, immunoglobulins, serum albumin, lactoferrin, and lactoperoxidase [2]. WPI also contains the peptide glycomacropeptide (GMP) [1].

Whey protein components have been characterized as immunomodulatory in in vitro cell models. For example, lactoferrin, a protein present in whey protein isolate, has an array of immunomodulatory activities [3]. Moreover, as whey protein is digested, it releases an array of peptides [4], including many with immunomodulatory activities [5]. For example, Ma et al. [6] found that alcalase-hydrolyzed WPI as a mixture (and specific peptide sequences found in the mixture) inhibited the gene expression of the inflammatory cytokines tumor necrosis factor-α (TNF-α) and interleukin-1β (IL-1β) in lipopolysaccharide (LPS)-stimulated mouse macrophages (RAW 264.7). Similarly, Silva et al. [7] found that whey protein hydrolysates inhibited the TNF-α gene expression of human endothelial cells stimulated with TNF-α.

Among the protein/peptide fraction of WPI, 20–25% is glycomacropeptide (GMP), making it the third most abundant protein/peptide after β-lactoglobulin and α-lactalbumin [8]. GMP is produced from the cleavage of k-casein by rennet between phenylalanine 105 and methionine 106 during cheesemaking [9]. GMP lacks aromatic amino acid residues and thus has been isolated as a protein source for people with phenylketonuria [10]. Purified GMP products consist of a mixture of glycosylated and non-glycosylated peptides [11]. A variety of glycans have been identified within the glycosylated form of GMP: disaccharides (*N*-acetyl galactosamine and galactose, GalNAcGal), trisaccharides (*N*-acetyl galactosamine, galactose, and *N*-acetyl neuraminic acid, GalNAcGalNeuAc, linear and branched), and tetrasaccharides (*N*-acetyl galactosamine, galactose, and two *N*-acetyl neuraminic acids, GalNAc_1_Gal_1_NeuAc_2_) [12]. Glycosylation can contribute up to 50% of GMP’s molecular weight [13]. These glycans increase GMP’s hydrophilicity and influence its bioactivity [13].

GMP has been shown to modulate various inflammatory pathways in different cell and animal models, with most of the literature indicating its anti-inflammatory effects within the innate immune response [14,15,16]. GMP appears to exert this anti-inflammatory effect by limiting activation of the nuclear factor-κB (NF-κB) and mitogen-activated protein kinase (MAPK) pathways [17]. MAPK and NF-κB pathways are common inflammatory signaling pathways active in the production of TNF-α, IL-1β, and IL-8 [18]. GMP binds to inflammatory LPS derived from pathogenic bacteria, preventing LPS’s adherence to cell receptors, which can limit inflammation [19]. GMP can also bind directly to bacteria, preventing their adhesion to epithelial cells, which can reduce resulting inflammation [20]. In contrast to the body of evidence that suggests GMP is anti-inflammatory, Requena et al. [17] observed that undigested GMP stimulated, rather than inhibited, inflammation. The authors observed GMP incubation resulted in dose-dependent increases in the inflammatory cytokines TNF-α, IL-1β, and IL-8 in both unstimulated Tohoku Hospital Pediatrics-1 (THP-1) monocytes and primary monocytes [17]. GMP also has performed a number of other bioactivities in cell and animal studies, including enhancing intestinal cell barrier function, modulating gut motility, and prebiotic activity [16].

The degree to which WPI and GMP are observed to exert immunomodulation in cell studies varies based on whether they were tested intact or partially digested and on the digestive enzymes used [20,21,22,23,24,25]. Peptides that are released from WPI and GMP during gastrointestinal digestion in the consumer would have the potential to affect immunoregulation by interacting with intestinal epithelial and immune cells. Koh et al. [26] demonstrated that during human consumption, GMP is digested into smaller peptides, many of which share homology with known bioactive peptides. The digestion-released GMP-derived fragments thus have a higher likelihood of being able to exert immunomodulatory effects within the human gut than the intact form of GMP. These initial investigations offer a rationale for further examining the immunomodulatory activity of WPI and GMP after digestion.

The immunomodulatory activities of WPI and GMP indicate their potential for use as dietary modulators of inflammatory diseases such as inflammatory bowel disease (IBD). Although the causes of IBD are not fully understood, specific immune cells and their inflammatory products, such as cytokines, have been implicated in IBD pathogenesis [27]. The pro-inflammatory cytokines TNF-α and IL-1β are higher in people with IBD than in healthy populations [28]. Serum concentrations of these cytokines, as well as interferon-γ (IFN-γ) and IL-6, are positively associated with worse IBD symptoms [28]. Experimental treatments for IBD, such as anti-TNF-α antibodies, have shown some success in targeting these cytokines and limiting IBD symptoms [29]. IL-1β has been shown to play a role in inducing T_h_17 cells, believed to be the primary effector cell in ulcerative colitis [28]. Conventional treatments for IBD typically involve the use of anti-inflammatory compounds like 5-aminosalicylic acid (5-ASA) or corticosteroid medications. However, not all patients achieve remission of symptoms, and some experience adverse side effects [30]. As alternative IBD treatments with fewer side effects are needed, investigation of immunomodulatory food components, such as WPI and GMP, is warranted.

Herein, we examine how digestion affects the degree to which WPI and GMP alter immunomodulatory activity in macrophage-like cells. We selected to study the effect of digested WPI and GMP on macrophage cells because macrophages play a central role in the immune system by actively phagocytizing foreign organisms, presenting antigens to T cells and releasing cytokines [31]. Macrophages are highly abundant in the human gut mucosa [32]. Gut macrophages can sample antigens from the intestinal lumen and initiate the immune response [33]. Understanding how GMP and WPI interact with macrophages will provide valuable insights into their potential immunomodulatory activity in the gastrointestinal tract.

## 2. Materials and Methods

### 2.1. Cell Type

We used macrophage-like cells that were derived through differentiation of the THP-1 monocytic cell line with phorbol 12-myristate 13-acetate (PMA). These cells represent a human-derived and immortalized model that responds to LPS stimulation via toll-like receptors, exhibiting a cytokine secretion profile similar to that of primary human macrophages [34]. Furthermore, the PMA-differentiated THP-1 cells share morphological characteristics and surface antigens with M1 subtype macrophages [35]. M1 macrophages can be activated by LPS to produce pro-inflammatory cytokines to initiate an immune response [36]. Given their well-established utility in the study of macrophage functions, PMA-differentiated THP-1 cells have been widely employed in previous research [37].

### 2.2. Samples

Five sample types were prepared for use in these experiments: (1) WPI; (2) in vitro-digested WPI (WPID); (3) GMP; (4) in vitro-digested GMP (GMPD); and (5) intestinally-digested WPI (WPIINT). All WPI used in these experiments was Provon 190 obtained from Glanbia Nutritionals (Fitchburg, WI, USA). Provon 190 is a non-fat whey protein isolate derived from sweet dairy whey, extracted using membrane filtration (>90% protein, <0.7% fat, <2% carbohydrate, <3.5% minerals, <5% moisture). Purified GMP, BiPRO GMP 9000, was procured from Agropur, Inc. (Le Sueur, MN, USA). This powder is composed of 74% protein, 96% of which is GMP. These samples, prepared using the following methods, will be referred to as follows: WPI, for undigested whey protein isolate; GMP, for undigested glycomacropeptide; WPID, for in vitro-digested WPI; GMPD, for in vitro-digested GMP; and WPIINT, for intestinally-digested WPI.

### 2.3. In Vitro Digestion

In vitro digestions (for WPID and GMPD) were carried out using a two-phase, static simulation of adult human digestion, as described previously [38]. Briefly, in the gastric phase, 0.75 g of WPI or GMP sample was added to 7.5 mL of simulated gastric fluid (SGF) electrolyte stock solution, 1.6 mL of 25,000 U/mL porcine pepsin stock solution made up in SGF electrolyte stock solution (pepsin from porcine gastric mucosa 3200–4500 U/mg protein, Sigma, St. Louis, MO, USA), 5 μL of 0.3 M CaCl_2_, 0.2 mL of 1 M HCl to reach pH 3.0, and 695 μL of water. The mixed liquid was incubated at 37 °C in a water bath for 2 h. In the intestinal phase, 20 mL of the simulated gastric chyme was mixed with 11 mL of simulated intestinal fluid (SIF) electrolyte stock solution, 5.0 mL of a 800 U/mL pancreatin solution made up in SIF electrolyte stock solution (pancreatin from porcine pancreas, Sigma, activity units based on trypsin activity), 2.5 mL bile salt solution (160 mM), 40 μL of 0.3 M CaCl_2_, 0.15 mL of 1 M NaOH to reach pH 7.0, and 1.31 mL of water. The mixed liquid was incubated at 37 °C in a water bath for 2 h. Herein, we refer to in vitro-digested WPI as WPID and in vitro-digested GMP as GMPD.

### 2.4. Feeding and Intestinal Sample Collection (In Vivo Digestion)

As described in our previous work, a nasopharyngeal tube was placed by trained nurses into the jejunum of a healthy adult male [39]. The subject then consumed a 1 L protein shake containing 1282 kcal, composed of 68 g of WPI (Provon 290, Glanbia Nutritionals, Twin Falls, ID, USA), 140 g of sucrose, and 150 mL of heavy whipping cream (50 g fat) in water, consumed by the subject within an hour. The subject also ingested an additional 2 L of water during that period. Jejunal fluids were collected up to 3 h post-protein shake consumption via gravity flow from the nasojejunal tube. As the digesta volume was collected, it was transferred to sterile 50 mL centrifuge tubes and kept on dry ice until it could be frozen at the end of sample collection. At the completion of the collection, this fluid was frozen at −80 °C prior to further experimentation. Herein, we refer to in vivo-digested WPI as WPIINT.

### 2.5. Peptide Extraction

Peptides were extracted from the in vitro- and intestinally digested samples prior to treatment of the THP-1 cells according to a previously published method [40]. This method applied ethanol precipitation to isolate peptides from proteins and C18 solid-phase extraction to isolate peptides from more polar substances (e.g., carbohydrates) and more nonpolar substances (e.g., lipids). Undigested GMP and WPI were not exposed to this treatment prior to testing.

### 2.6. Cell Culture

The THP-1 macrophage cell line (American Type Culture Collection) was cultured and maintained in 25 cm^2^ culture flasks using RPMI 1640 medium. The growth medium consisted of 10% (*v*/*v*) filter-sterilized fetal bovine serum (FBS, Sigma, St. Louis, MO, USA) and 0.05 mM 2-mercaptoethanol (Sigma). The flasks were incubated in a humidified incubator with 5% CO_2_ at 37 °C. The growth medium was refreshed every two days. Once the cell concentration reached to 1 × 10^6^ cells/mL, the THP-1 cells were ready for seeding. For experiments, the THP-1 cells were seeded at a concentration of 5 × 10^5^ cells/mL in 24-well tissue culture plates and subjected to differentiation. This differentiation process involved the use of a medium containing 100 ng/mL phorbol 12-myristate 13-acetate (PMA) for a duration of three days, following a previously established protocol [40]. After the three-day PMA treatment, cells were washed with phosphate-buffered saline (PBS) three times and further incubated with fresh RPMI 1640 supplemented with 10% FBS and 0.05 mM 2-mercaptoethanol. The cells were maintained in this medium for an additional five days, with regular medium changes every two days. Subsequently, the cells were considered ready for experimental use.

### 2.7. Cell Viability Assay

To ensure the viability of the cells when exposed to the sample peptides, an MTT (3-[4,5-dimethylthiazol-2-yl]-2,5-diphenyltetrazolium bromide) assay was completed. This assay was completed for all treatments: the control containing 10 ng/mL LPS (derived from *Escherichia coli* O111:B4, Millipore Sigma, Burlington, VT, USA), the treatment wells containing peptides and 10 ng/mL LPS, and the blanks containing only growth media. This assay indirectly tests the viability of cells by evaluating their metabolic activity [41]; living cells will convert MTT into formazan, which has a purple color that can be quantified by measuring changes in absorbance at 540 nm.

Cells were rinsed with 100 µL of PBS. Peptides were dissolved in RPMI medium containing 1% FBS and 0.05 mM 2-mercaptoethanol. The prepared peptide solutions were then filter-sterilized using 0.2 µm syringe filters. Two milliliters of each sample type, which included the control, blank, and peptides at a concentration of 1000 µg/mL were introduced into the wells of the culture plates. These plates were then incubated for 24 h in humidified incubators with 5% CO_2_ at 37 °C. Following the treatment period, the cells were rinsed with PBS and incubated for an additional 24 h with serum-free medium that contained 0.5 mg/mL MTT. Plates were incubated in the dark at 37 °C. Ten percent *w*/*v* sodium dodecyl sulfate in 0.1 M hydrochloric acid was added to the cells for 12 h to solubilize formazan. Formazan was quantified using a spectrophotometer measuring absorbance at 540 nm. Viability was calculated using the following equation: viability (% of control) = (Ab treatment/Ab negative control) × 100%. All the cell viability experiments were conducted on three batches of cells (on three different days). In each batch, each treatment was tested in duplicate wells.

### 2.8. Immunomodulatory Assay

Peptides were prepared in a growth medium as described above. The THP-1-derived macrophages were pre-treated with individual peptide samples for 24 h. Peptides/proteins were tested at 10, 100, and 1000 µg/mL final concentrations in the medium.

The medium was replaced with fresh peptide/protein samples in the same medium at the same concentrations with the addition of 10 ng/mL *Escherichia coli* (0111:B4) LPS as an inflammatory challenge and incubated for 20 h. Cells not exposed to peptide but with LPS challenge served as the positive control. The blanks were cells with neither peptide treatment nor LPS challenge.

Cells were incubated overnight, and the medium was collected into microcentrifuge tubes prior to analysis. Immunomodulatory effects of the extracted peptides/proteins were determined by measuring alterations in the macrophages’ production of the cytokines IL-1β and TNF-⍺ as measured by ELISA. All the cell culture experiments were conducted on three batches of cells (on three different days). In each batch, each treatment was tested in duplicate wells.

### 2.9. Quantification of TNF-α and IL-1β by Enzyme-Linked Immunosorbent Assay (ELISA)

To evaluate the inflammatory response of the cells, the cytokines TNF-α and IL-1β were analyzed using ELISA. ELISA kits (TNF alpha Human ELISA Kit, BMS223-4TEN, IL-1 beta Human ELISA Kit, BMS224-2TEN) were purchased from ThermoFisher Scientific (Waltham, MA, USA). ELISAs were carried out manually according to the manufacturer’s instructions. Human cytokine ELISA kits were selected because THP-1 is a human-derived cell line. All samples were analyzed in duplicate.

### 2.10. Statistics

Prior to statistical analyses, duplicate ELISA concentration measurements were averaged. Tests were performed with *n* = 6 technical replicates (3 days of repeated cell culture experiments with each sample tested in duplicate wells). Two-sided *t*-tests were performed to determine whether each cytokine concentration (TNF-α and IL-1β) in each treatment (WPI, GMP, WPID, GMPD, WPIINT at each dose (10, 100, and 1000 µg/mL)) were significantly different from the LPS-only control using GraphPad Prism software (version 8.2). The significance of the differences in percentage change of each cytokine (TNF-⍺ and IL-1β) from the LPS-only control among WPI, WPID, and WPIINT at each concentration, and among GMP and GMPD at each concentration, were assessed using multi-factor ANOVA followed by Tukey’s HSD tests, using JMP software (version 17). Significance for both statistical tests was defined as *p* < 0.05.

## 3. Results

### 3.1. Toxicity Assay

The results from the toxicity assay indicate that the experimental treatments used on these cells did not compromise their viability (>90% viability) (Appendix A).

### 3.2. Undigested WPI Did Not Impact TNF-α but Decreased IL-1β Production

Undigested WPI had no effect on TNF-α production at any concentration (Figure 1). Data were expressed as a percentage of control. Mean concentrations are presented in Appendix A. Undigested WPI decreased IL-1β at all concentrations compared to the LPS-only control (Figure 2, Appendix A). IL-1β expression was reduced by 55 ± 10%, 68 ± 6%, and 74 ± 7%, at concentrations 10 µg/mL, 100 µg/mL, and 1000 µg/mL, respectively (Table 1). The 1000 µg/mL dose of undigested WPI caused significantly greater reductions in IL-1β than the 10 µg/mL dose, indicating a dose dependency.

### 3.3. In Vitro-Digested WPI Decreased TNF-α and IL-1β Production

In vitro-digested WPI decreased TNF-α at all concentrations (Figure 1). This treatment appears to have had a dose-dependent effect, with the 10 µg/mL, 100 µg/mL, and 1000 µg/mL treatment doses resulting in decreases in TNF-α concentration by 21 ± 10%, 37 ± 10%, and 55 ± 8%, respectively, compared to the control (Table 1). The 1000 µg/mL dose of in vitro-digested WPI caused significantly greater reductions in TNF-α than the 10 µg/mL dose (Appendix A).

In vitro-digested WPI decreased IL-1β production at all concentrations (Figure 2). Each of the treatment doses produced a similar degree of inhibition for IL-1β (no significant differences, Appendix A). Even the lowest dose at 10 µg/mL reduced IL-1β concentration by 57% ± 10%. The 100 µg/mL and 1000 µg/mL doses reduced IL-1β concentration by 63 ± 8% and 65 ± 6%, respectively (Table 1).

### 3.4. Intestinally Digested WPI Decreased TNF-α and Caused Dose-Dependent Changes in IL-1β Production

Intestinal digested WPI significantly decreased TNF-α at all concentrations (Figure 1). Of the treatments tested, the intestinally digested WPI had the greatest inhibition of TNF-α (Figure 1). Ten µg/mL, 100 µg/mL, and 1000 µg/mL peptide concentrations decreased TNF-α concentrations by 84 ± 3%, 85 ± 12%, and 86 ± 11%, respectively (Table 1) (no significant differences).

Unlike the results for TNF-α, the intestinal digests of WPI decreased IL-1β at 10 µg/mL by 62 ± 7%, had no effect at 100 µg/mL, and increased IL-1β expression at 1000 µg/mL by 44 ± 21% (Figure 2, Table 1). The change in IL-1β was significantly different at each dose.

### 3.5. Undigested GMP Increased TNF-α and IL-1β Production

Undigested GMP increased production of TNF-α at 100 µg/mL and 1000 µg/mL (Figure 3). The higher doses, 100 µg/mL and 1000 µg/mL, increased TNF-α concentration by 26 ± 20% and 33 ± 17%, respectively (Figure 3, Table 1).

Similarly, undigested GMP increased production of IL-1β at 100 µg/mL and 1000 µg/mL (Figure 4). At higher doses, undigested GMP increased IL-1β concentration (by 47 ± 24% and 73 ± 32% at 100 µg/mL and 1000 µg/mL, respectively). The change in IL-1β was significantly different at each dose (increasing cytokine production with increasing undigested GMP concentration).

### 3.6. In Vitro-Digested GMP Decreased TNF-α and IL-1β Production

In vitro-digested GMP decreased TNF-α at all concentrations (Figure 3). In vitro-digested GMP at 10, 100, and 1000 µg/mL reduced TNF-α by 32 ± 6%, 58 ± 3%, and 69 ± 7%, respectively (Table 1). Treatments at 100 and 1000 µg/mL were not statistically different from each other but did differ from the 10 µg/mL treatment (Appendix A). 

Treatment with in vitro-digested GMP also reduced the expression of IL-1β at all concentrations (Figure 4). The in vitro-digested GMP reduced IL-1β expression by 41 ± 11%, 81 ± 8%, and 66 ± 6% at 10 µg/mL, 100 µg/mL, and 1000 µg/mL, respectively (Figure 4, Table 1). The higher treatment concentrations were not statistically different from each other but did differ from the 10 µg/mL treatment (Appendix A). 

## 4. Discussion

Our findings suggest that both digestion state and concentration contribute to the effects of WPI and GMP on TNF-α and IL-1β secretion in LPS-challenged THP-1 macrophages.

We observed that undigested WPI did not significantly impact TNF-α expression. Although many in vitro [6,7,24] and mouse studies [42,43] demonstrate that WPI has immunomodulatory properties, including inhibiting TNF-α expression or production; in most cases, this activity is demonstrated when the WPI is pre-hydrolyzed in cell studies or digested in the gut in animal studies. Our study suggests that the release of bioactive peptides is important for WPI’s impact on pathways that modulate TNF-α.

In contrast to its effect on TNF-α, undigested WPI reduced IL-1β expression at all tested concentrations, in a dose-dependent manner. To our knowledge, this effect of WPI on IL-1β has not been previously observed in THP-1 cells.

In vitro-digested WPI decreased TNF-α expression in a dose-dependent manner. These findings align with the current body of literature indicating that digested WPI has anti-inflammatory properties. Previous studies have reported that in vitro-digested WPI reduced inflammation, as indicated by decreased production of TNF-α [7] and inhibition of the NF-kB pathway [44]. These studies suggest that the bioactive components released during the digestion process of WPI have anti-inflammatory effects, particularly in modulating TNF-α production.

Similarly, in vitro-digested WPI significantly inhibited IL-1β production. This finding aligns with that of Jayatilake et al. [45], who found that mice fed WPI had reduced markers of dextran sodium sulfate-induced colonic damage and inflammation.

We observed that intestinally digested WPI had the most significant inhibitory effect on TNF-α expression among all the treatments tested. This potent inhibition of TNF-α production is significant, given the central role of TNF-α in various inflammatory processes.

In contrast to the results for TNF-α, the effects of intestinally digested WPI on IL-1β expression were highly concentration dependent, with the lowest dose (10 µg/mL) decreasing IL-1β production, the middle dose (100 µg/mL) having no effect on IL-1β, and the highest dose (1000 µg/mL) increasing IL-1β expression. Further research examining the underlying mechanisms of this observation are needed. Likely, this response is the result of multiple compounds in the sample having interacting effects on pathways that control IL-1β expression.

The difference in results for intestinally digested WPI and in vitro-digested WPI may suggest that the in vitro digestion inadequately mimicked adult human digestion, resulting in different protein/peptide digestive profiles with different overall bioactivity. The observed difference in results could also be due to the presence of other compounds in the intestinally digested WPI secreted into the intestine by the subject prior to sample collection.

We observed that undigested GMP increased the production of both TNF-α and IL-1β in LPS-challenged macrophages. These effects were dose-dependent, with higher concentrations of undigested GMP resulting in increased production of these pro-inflammatory cytokines. These findings are unexpected, given the breadth of evidence that indicates that GMP has anti-inflammatory properties. However, undigested GMP has been shown previously to behave differently than its digested counterpart. Requena et al. [17] found similar increases in inflammatory cytokines when testing undigested GMP in unstimulated THP1 monocytes. Requena et al. [17] suggest that undigested GMP does not bind to LPS and may impact other inflammatory pathways, including NF-κB and MAPK.

Although intact GMP increased production of both inflammatory cytokines, this undigested form is unlikely to reach intestinal macrophages. When fed, GMP undergoes digestion within the gastrointestinal tract. Koh et al. [26] demonstrated that GMP is broken down into many peptide fragments during human digestion, including many that are known bioactive peptides. We observed that in vitro-digested GMP strongly inhibited production of both TNF-α and IL-1β. These findings align with previous research, which has suggested that digested forms of GMP possesses immunomodulatory properties. For example, Sawin et al. [46] found that mice fed GMP had reduced blood plasma levels of TNF-α and IL-1β compared with mice fed casein. Although Daddaoua et al. [47] did not directly measure TNF-α and IL-1β, they observed that feeding GMP reduced colonic damage in a hapten-induced colitis mouse model. Not all animal feeding studies indicate anti-inflammatory actions of GMP in its digested form: for example, Ortega-Gonzalez et al. [48] found that GMP supplementation in rats with dextran sulfate sodium-induced colitis increased TNF-α in mesenteric lymph node cells. Although not a direct measure of cytokines, Li et al. [49] found that in vitro-digested GMP increased the cell proliferation and phagocytic activities of monocytes.

Many of our findings revealed clear dose-dependent effects on TNF-α and IL-1β secretion. However, some of the results did not yield a clear, consistent dose-dependent effect. In some cases, the mean values represent dose-dependent effects, but the values are not statistically different with increasing doses. Repeating the experiment to add additional experimental replicates would likely help identify additional statistical differences. Some results exhibit a dose-dependent effect at concentrations of 10 and 100 μg/mL (GMPD on TNF-α and IL-1β secretion) but not at 1000 μg/mL. These findings indicate a possible saturation threshold effect. To identify more dose-dependent responses, further investigations can include additional sample concentrations and increased replicates.

Overall, these results indicate that digestion modulates the immunomodulatory actions of both GMP and WPI. The digestion process releases an array of peptides from GMP and WPI [8]. Many milk protein-derived peptides have known bioactivities [26].

The ability of in vitro-digested GMP and WPI to reduce the expression of pro-inflammatory cytokines, particularly TNF-α and IL-1β, has potential clinical implications for inflammation-related conditions. Further research is warranted to elucidate the underlying mechanisms through which in vitro-digested WPI and GMP exert these anti-inflammatory effects.

WPI appears to have anti-inflammatory effects in human feeding studies. A recent meta-analysis found that whey protein consumption led to a reduction in circulating IL-6 levels [50]. Though GMP has immunomodulatory effects in cell and animal studies, human trials assessing the impact of GMP consumption on inflammation found little effect [21,51]. Further human clinical research is necessary to validate findings in cell and animal models.

## 5. Conclusions

Our study indicates that digestion of GMP and WPI gives rise to anti-inflammatory effects in macrophages. Given the evidence generated during this study, WPI and GMP have promise as a potential aid in reducing inflammation in the gastrointestinal tract, which could be useful in conditions such as IBD. Further cellular and human studies are necessary to understand the effects of GMP and WPI on the gut immune system.

## Figures and Tables

**Figure 1 nutrients-15-04942-f001:**
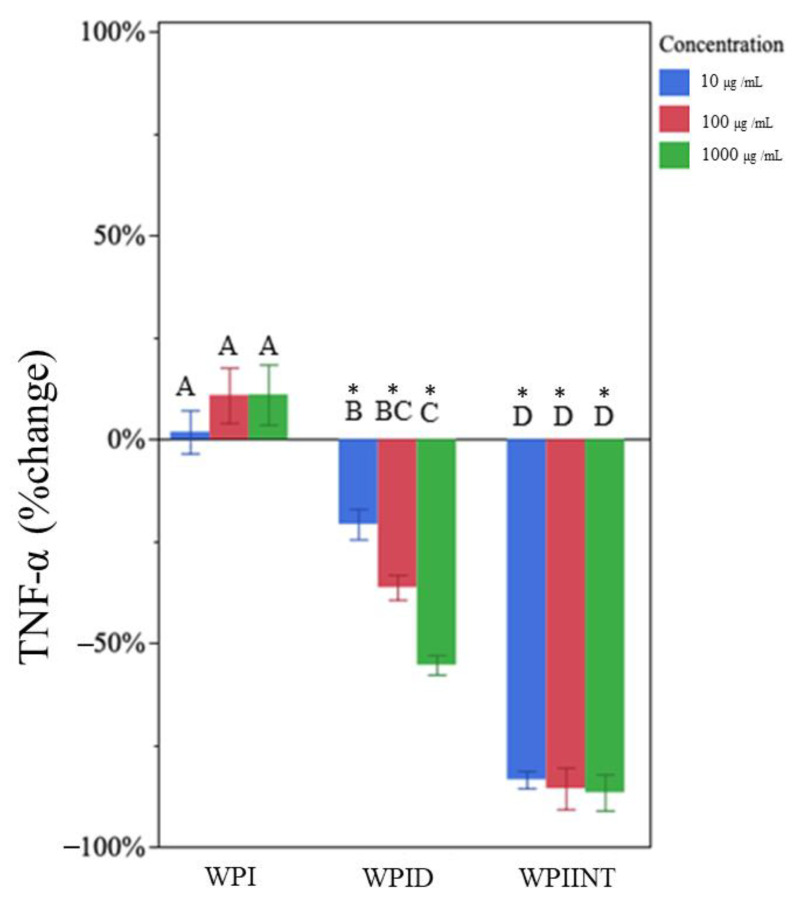
Percentage change in TNF-α concentration for each whey protein isolate sample type (whey protein isolate (WPI), in vitro-digested WPI (WPID), and intestinally digested WPI (WPIINT) at each concentration level) compared to the LPS-only control. Each treatment was administered at 10 (blue), 100 (red), and 1000 (green) µg/mL in medium. Bar heights represent mean values, with brackets representing ± standard deviations (n = 6 technical replicates (3 days of cell experiment replication, duplicate wells on each day)). Stars above each bar indicate statistically significant differences from the LPS-only control based on the two-sided *t*-test (*p* < 0.05). Letters indicate the significance of differences in percent change of TNF-⍺ from the LPS-only control among WPI, WPID, and WPIINT at each concentration based on a multi-factor ANOVA followed by Tukey’s HSD tests.

**Figure 2 nutrients-15-04942-f002:**
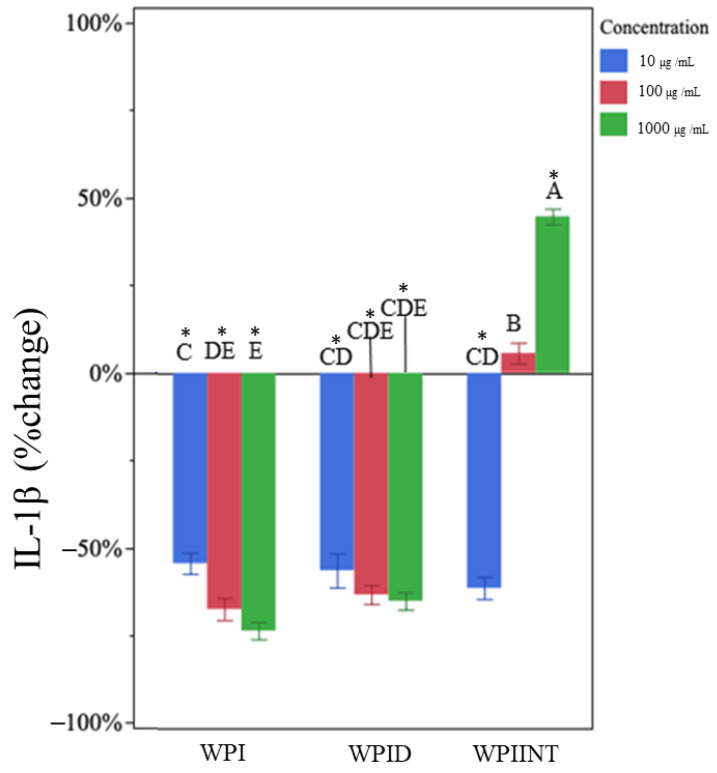
Percentage change in IL-1β concentration for each whey protein isolate sample type (whey protein isolate (WPI), in vitro-digested WPI (WPID), and intestinally digested WPI (WPIINT) at each concentration level) compared with the LPS-only control. Each treatment was administered at 10 (blue), 100 (red), and 1000 (green) µg/mL in medium. Bar heights represent mean values, with brackets representing ± standard deviations (n = 6 technical replicates (3 days of cell experiment replication, duplicate wells on each day)). Stars above each bar indicate statistically significant differences from the LPS-only control based on the two-sided *t*-test (*p* < 0.05). Letters indicate the significance of differences in percentage change of IL-1β from the LPS-only control among WPI, WPID, and WPIINT at each concentration based on a multi-factor ANOVA followed by Tukey’s HSD tests.

**Figure 3 nutrients-15-04942-f003:**
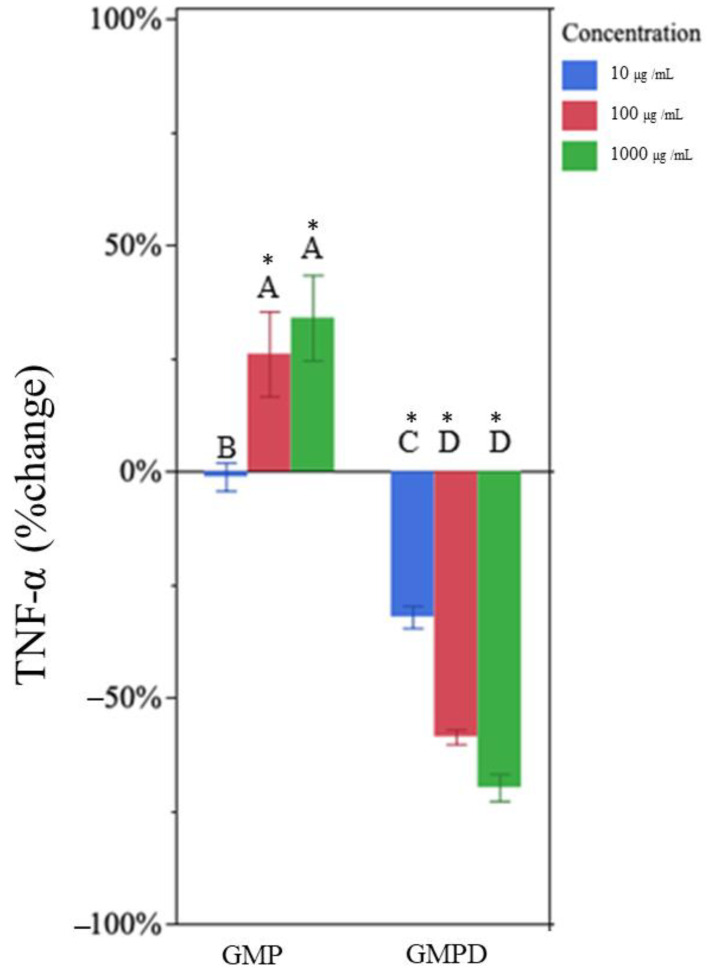
Percentage change in TNF-α concentration for each glycomacropeptide sample type (Glycomacropeptide (GMP), in vitro-digested GMP (GMPD), at each concentration) compared to the LPS-only control. Each treatment was administered at 10 (blue), 100 (red), and 1000 (green) µg/mL in medium. Bar heights represent mean values, with brackets representing ± standard deviations (n = 6 technical replicates (3 days of cell experiment replication, duplicate wells on each day)). Stars above each bar indicate statistically significant differences from the LPS-only control based on the two-sided *t*-test (*p* < 0.05). Letters indicate the significance of differences in percentage change of TNF-⍺ from the LPS-only control among GMP and GMPD at each concentration based on a multi-factor ANOVA followed by Tukey’s HSD tests.

**Figure 4 nutrients-15-04942-f004:**
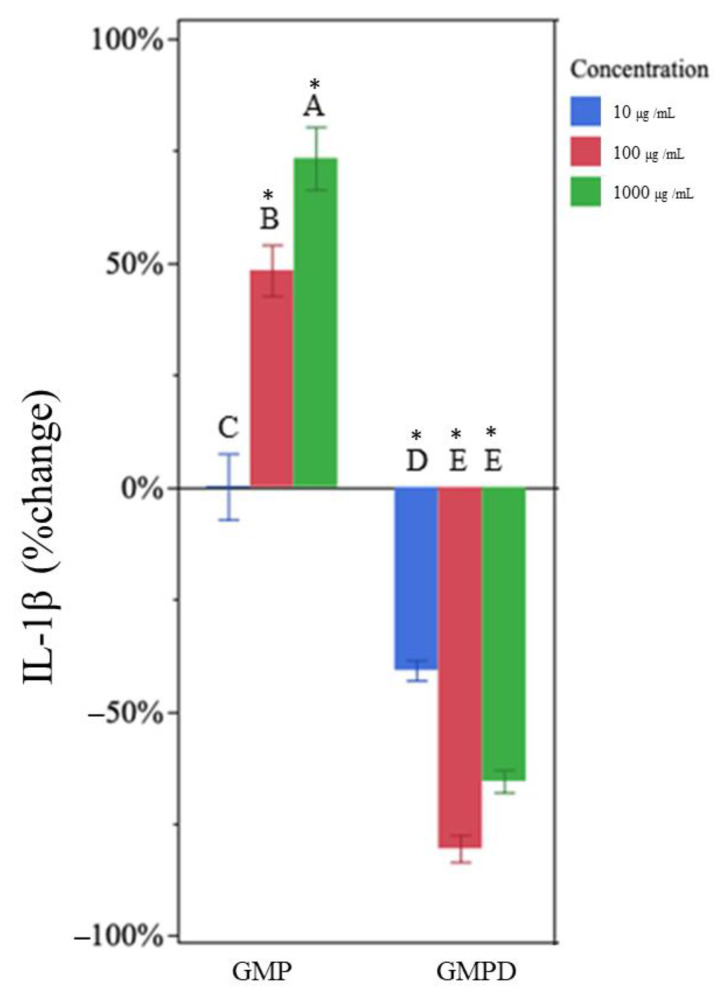
Percentage change in IL-1β concentration for each glycomacropeptide sample type (glycomacropeptide [GMP], in vitro-digested GMP [GMPD], at each concentration) compared to the LPS-only control. Each treatment was administered at 10 (blue), 100 (red), and 1000 (green) µg/mL in medium. Bar heights represent mean values, with brackets representing ± standard deviations (n = 6 technical replicates (3 days of cell experiment replication, duplicate wells on each day)). Stars above each bar indicate statistically significant differences from the LPS-only control based on the two-sided *t*-test (*p* < 0.05). Letters indicate the significance of differences in percentage change of IL-1β from the LPS-only control among GMP and GMPD at each concentration based on a multi-factor ANOVA followed by Tukey’s HSD tests.

**Table 1 nutrients-15-04942-t001:** TNF-α and IL-1β percentage difference from LPS-only control and standard deviation (SD).

Treatment Outcomes					
		Percentage Difference			
Treatment (µg/mL)		TNF-α		IL-1β	
		Avg. Diff.	SD	Avg. Diff.	SD
WPI ^a,b^	10	1%	6%	−55% *	10%
	100	10%	10%	−68% *	6%
	1000	10%	7%	−74% *	7%
WPID ^c^	10	−21% *	10%	−57% *	10%
	100	−37% *	8%	−63% *	8%
	1000	−55% *	7%	−65% *	6%
GMP ^d^	10	−1%	9%	−1%	18%
	100	26% *	20%	47% *	24%
	1000	33% *	17%	73% *	32%
GMPD ^e^	10	−32% *	6%	−41% *	11%
	100	−58% *	3%	−81% *	8%
	1000	−69% *	7%	−66% *	6%
WPIINT ^f^	10	−84% *	3%	−62% *	7%
	100	−85% *	12%	6%	18%
	1000	−86% *	11%	44% *	21%

^a^ Values statistically different from the LPS-only control. One star (*) indicates *p* < 0.05; n = 6 technical replicates (3 days of cell experiment replication, duplicate wells on each day)). ^b^ Whey protein isolate (WPI). ^c^ In vitro-digested whey protein isolate (WPID). ^d^ Glycomacropeptide (GMP). ^e^ In vitro-digested glycomacropeptide (GMPD). ^f^ Intestinally-digested whey protein isolate (WPIINT).

## Data Availability

Data are contained within the article and Appendix A.

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
