# Peer review of "Macrophage-Immunomodulatory Actions of Bovine Whey Protein Isolate, Glycomacropeptide, and Their In Vitro and In Vivo Digests"

_nutrients, 2023, doi:10.3390/nu15234942_

Round 1

Reviewer 1 Report

Comments and Suggestions for Authors

Congratulations to the authors of a very interesting article. A very well-prepared, clear introduction referring to previous research. Results presented correctly and appropriately described in the discussion.

I have only a few comments regarding the "material and methods" part:

1. Please let me know how many repetitions a given analysis/cell culture was performed? I did not find such information in the text.

2. The Elisa method is very sensitive to the type of execution. Please let me know whether the tests were performed manually or automatically? How many repetitions was the sample analyzed? Please complete the text.

Author Response

Comments and Suggestions for Authors

Congratulations to the authors of a very interesting article. A very well-prepared, clear introduction referring to previous research. Results presented correctly and appropriately described in the discussion.

I have only a few comments regarding the "material and methods" part:

  1. Please let me know how many repetitions a given analysis/cell culture was performed? I did not find such information in the text.

A1: All the cell culture experiments were conducted on three batches of cells. In each batch, each treatment was conducted in duplicate wells. This information has been added to lines 201-202.

  1. The ELISA method is very sensitive to the type of execution. Please let me know whether the tests were performed manually or automatically? How many repetitions was the sample analyzed? Please complete the text.

A2: All the ELISA steps were performed manually. All the supernatant samples were analyzed in duplicate. We have added this information to lines 208-210.

Reviewer 2 Report

Comments and Suggestions for Authors

I would like to offer some points of clarification and direct a few inquiries to the authors:

1. In the entirety of the manuscript, please adhere to academic writing conventions for the abbreviation or expansion of terms, for example in lines 15, 19, and 46 for IBD, LPS, THP, and DQWL.

2. In the introduction section, the description of the active components and immunomodulatory effects of WPI appears disorganized, despite the stratified writing approach already in place. It is recommended that this section be reorganized. 

3. Please include a more detailed analysis of the components of WPI, complemented by tables.

4. A clarification is required on the choice of utilizing human-sourced, instead of animal-sourced, cytokines for the ELISA analysis. 

5. The production processes for WPID and WPIINT need to be explicated in greater detail.

6. The manuscript consistently presents cytokine content in percentages, as opposed to actual measured amounts. This methodological choice requires justification.

7. The figure captions lack detailed statistical annotations and explanations.

8. There is an absence of statistical results in the supplementary material.

9. The experiments were conducted using varying concentrations, yet the results do not exhibit a dose-dependent effect. This observation requires a thorough explanation in the discussion section.

10. The manuscript solely analyzes cytokines and lacks other immune indices or results on the suppression of mediated immune pathways. Further validation is required, at the very least including results from Western blot analyses.

11. Experiments comparing the results with RAW 264.7 cells should be conducted to ascertain whether the conclusions remain consistent.

Comments on the Quality of English Language

I kindly urge adherence to academic English writing conventions.

Author Response

Reviewer 2: 

Comments and Suggestions for Authors

I would like to offer some points of clarification and direct a few inquiries to the authors:

  1. In the entirety of the manuscript, please adhere to academic writing conventions for the abbreviation or expansion of terms, for example in lines 15, 19, and 46 for IBD, LPS, THP, and DQWL.

A1: Thank you for the suggestion. We have reviewed the text to ensure that the full name of each abbreviation is provided first, followed by the abbreviation in parentheses and then use of the abbreviation in all following cases.

  1. In the introduction section, the description of the active components and immunomodulatory effects of WPI appears disorganized, despite the stratified writing approach already in place. It is recommended that this section be reorganized. 

A2: The paragraph describing the active components and immunomodulatory effects of WPI has been reorganized. Please refer to line 38-48.

  1. Please include a more detailed analysis of the components of WPI, complemented by tables.

A3: We have added additional information about the WPI and GMP products in the methods section. Lines 125-137.

  1. A clarification is required on the choice of utilizing human-sourced, instead of animal-sourced, cytokines for the ELISA analysis. 

A4: The THP1-derived macrophage is a human sourced cell line. Therefore, the cytokines that it produces will have the human amino acid sequences and structure. Therefore, a human-specific cytokine ELISA was selected. This justification was added to line 230-231.

  1. The production processes for WPID and WPIINT need to be explicated in greater detail.

A5: Thank you for the constructive feedback. We added detailed information about the in vitro digestion process for WPID and the in vivo digestion process for WPIINT. Please refer to line 125-137 for in vitro digestion and line 153-163 for the in vivo digestion.

  1. The manuscript consistently presents cytokine content in percentages, as opposed to actual measured amounts. This methodological choice requires justification.

A6: We expressed the cytokine content in percentages because we wanted the graphical representations to be consistent with the way we discuss the changes induced by the peptides in the text, in which we refer to % changes. The absolute values of the cytokines were deemed less important than the percent change in cytokine expression. We have added the actual values to the supplementary table 2.

  1. The figure captions lack detailed statistical annotations and explanations.

A7: In each of the figures, we have explained the definition of the stars and the lettering based on the specific statistical tests used. We have also explained these tests in the Methods section.  We added additional information describing that the bar heights represent the mean values with brackets representing ± standard deviations (3 days of replication, duplicates on each day, with values measured in duplicate). We clarified how replicates were used in the statistical analysis section (lines 234-236).

  1. There is an absence of statistical results in the supplementary material.

A8: We have added our statistical results to the supplementary material.

  1. The experiments were conducted using varying concentrations, yet the results do not exhibit a dose-dependent effect. This observation requires a thorough explanation in the discussion section.

A9: Thank you for the suggestion. Though dose-dependent effects are present in most of our data, some data do not show a dose-dependent effect. We added a greater clarity on dose-dependency in the results and throughout the Discussion section and a specific paragraph about dose-dependent effects in the Discussion section (lines 413-430).

  1. The manuscript solely analyzes cytokines and lacks other immune indices or results on the suppression of mediated immune pathways. Further validation is required, at the very least including results from Western blot analyses.

A10: Thank you for your thorough review of our manuscript, and we appreciate your insightful comments. We carefully considered your suggestion to conduct additional experiments, including performing western blot analyses to further validate protein expression levels of key markers involved in the immune pathways. While we value your input, we would like to provide our rationale for not pursuing further experiments at this stage.

First, it is our opinion that ELISA results for cytokine expression levels are more valuable than Western blotting for these cytokines as ELISA provides a much more accurate quantitative analysis. We believe that Western blotting would provide little additional information.

Second, our manuscript was designed with a specific research question which is how digestion impact immunomodulatory activity of WPI and GMP in human immune cells. The study's scope and objectives were intentionally defined to address this aspect of the immune response. Expanding the scope to mechanisms of immunomodulatory action could be our next research question in our future studies. While we acknowledge the potential benefits of an in-depth mechanisms study, we believe that the current findings and the focus on cytokines provide valuable insights into the specific aspect of the immune response under investigation. These findings are a valuable contribution to the field, and it is our hope that they will stimulate further research and discussions in this area. We will, of course, take your feedback into account in our discussion and conclusion sections, highlighting the limitations of our study and suggesting areas for further research.

  1. Experiments comparing the results with RAW 264.7 cells should be conducted to ascertain whether the conclusions remain consistent.

A11: Thank you for the idea. Herein, we selected to focus on how digestion affects the degree to which WPI and GMP alter immunomodulatory activity in human macrophage-like cells. Therefore, we chose a human macrophage model instead of mouse macrophage model (like RAW 264.7). THP-1 derived macrophages are a type of in vitro cultured macrophages that originate from the human monocytic cell line THP-1. THP-1 cells are typically cultured in a suspension medium in the undifferentiated state. However, they can be induced to differentiate into macrophage-like cells by various stimuli, commonly with phorbol 12-myristate 13-acetate (PMA), a compound that activates protein kinase C. When treated with PMA, THP-1 cells undergo morphological changes, adhering to culture plates and acquiring characteristics of macrophages. These differentiated cells become adherent, exhibit increased phagocytic activity, and express cell surface markers and receptors typically found on macrophages. Future work can examine whether these effects are similar with mouse origin RAW 264.7 cells. While they provide valuable insights into immune responses and inflammation in mice, there can be variations in how mouse cells respond compared to human cells. This limits their direct translatability to human physiology, which may be a concern in our research contexts.

Thank you for your consideration. We look forward to hearing from you.

Round 2

Reviewer 2 Report

Comments and Suggestions for Authors

The revised manuscript has been improved a lot. Minor comments on adding the N values in supplementary data.

Comments on the Quality of English Language

None.

Author Response

Thank you for your comments. We have added the N values to all figures and tables in the supplementary data.